# Ferroptosis in Autoimmune Diseases: Research Advances and Therapeutic Strategies

**DOI:** 10.3390/ijms262110449

**Published:** 2025-10-28

**Authors:** Ziman He, Bo Liu, Zuquan Xian, Aimin Gong, Xiaokang Jia

**Affiliations:** 1School of Traditional Chinese Medicine, Hainan Academy of Medical Sciences, Hainan Medical University, Haikou 571199, China; 18620574571@163.com (Z.H.); xzq13337698750@163.com (Z.X.); 2Pharmacology Laboratory, Zunyi Medical University, Guizhou 563000, China; yaoliliubo@zmu.edu.cn; 3Traditional Chinese Medicine Diagnosis Teaching and Research Office, College of Basic Medicine, Zhejiang Chinese Medical University, Hangzhou 310053, China

**Keywords:** ferroptosis, autoimmune diseases, JAK/STAT pathway, NF-κB pathway, lipid peroxidation, precision therapy

## Abstract

Ferroptosis, an iron-dependent programmed cell death driven by lipid peroxidation, plays a critical role in autoimmune diseases such as rheumatoid arthritis, systemic lupus erythematosus, and psoriasis. This review systematically explores the interaction between ferroptosis and the immune system, highlighting its dynamic regulation of immune cell function (e.g., Treg cell stability, neutrophil activity) and inflammatory microenvironments via signaling pathways including JAK/STAT and NF-κB. Ferroptosis suppresses inflammation in rheumatoid arthritis by eliminating pro-inflammatory synoviocytes but exacerbates tissue damage in systemic lupus erythematosus through neutrophil ferroptosis. While ferroptosis inhibitors (e.g., Fer-1) and inducers (e.g., IKE) show promise in preclinical models, clinical translation faces challenges such as disease-specific mechanistic heterogeneity, insufficient drug selectivity, and complex metabolic interactions. Future research should integrate multi-omics, organoid models, and AI-driven predictions to develop precision-targeted strategies, offering novel therapeutic paradigms for autoimmune diseases.

## 1. Introduction

The immune system uses a variety of immune cells to eliminate senescent cells and immune complexes in the body and establish autoimmune tolerance to resist foreign invading pathogens, which is a defense mechanism of the human body. In certain instances, the disruption of autoimmune tolerance can lead to an aberrant immune response where the body’s immune system attacks its own tissues and cells, causing cellular or tissue damage and associated clinical manifestations, ultimately resulting in autoimmune diseases [1,2,3].

Approximately 20% of the global population is impacted by over 100 distinct types of autoimmune diseases. Among them, psoriasis (2–4%), rheumatoid arthritis (0.5–1%), Graves’ disease (0.5%), Crohn’s disease (0.2–0.3%) and systemic involvement of systemic lupus erythematosus were the most common [4]. As early as 1999, the World Health Organization listed autoimmune diseases as the third major threat to human health after cardiovascular diseases and cancer. After summarizing the different prevalence of Acquired Immune Deficiency Syndrome (AIDs) in 11 countries, T Ngo found that the incidence of AIDs in women was higher than that in men [5]. In the past few decades, the overall incidence of AIDs has shown an upward trend, which has brought harm to people’s life and economy [3,6]. At present, the research on AIDs is multi-faceted, involving a variety of pathways and microstates. Among them, ferroptosis is a more popular research direction. Ferroptosis is a new type of programmed cell death different from apoptosis, which has the characteristics of immunogenic cell death. Ferroptosis can be caused by abnormal lipid accumulation in immune cells, which in turn can act on abnormal expression of immune cell function [7]. Ferroptosis has been proved to play a role in different diseases, and previous studies have also carried out experimental analysis and explanation of the mechanism of ferroptosis in the immune system from different angles. In this paper, a comprehensive review of multiple studies was conducted to examine the role of ferroptosis in the immune system.

## 2. Overview and Important Components of Ferroptosis

### 2.1. Overview

The concept of ferroptosis is a form of programmed cell death (PCD) that is different from apoptosis and autophagy, which was first proposed by Dixon et al. in 2012 [8]. Cellular features of ferroptosis include loss of membrane integrity, increased membrane density, mitochondrial shrinkage, and rupture of the mitochondrial outer membrane, but normal nuclear morphology [8]. Current studies suggest that ferroptosis is related to fatal lipid peroxidation, and ferroptosis is also the result of imbalance of cellular metabolism and redox homeostasis. After literature review, we found that ferroptosis plays an important physiological role in the occurrence of neurodegenerative diseases, ischemic organ damage [9], gastrointestinal system diseases, tumors, and immune diseases.

### 2.2. Important Components and Pathways of Ferroptosis

Ferroptosis is an iron-dependent form of regulated cell death characterized by unchecked lipid peroxidation. Two central regulatory systems are the glutathione peroxidase 4 (GPX4) axis and the system Xc^−^ pathway. GPX4, a selenoprotein, serves as the primary antioxidant enzyme that reduces lipid hydroperoxides (L-OOH) to their corresponding alcohols, thereby protecting cell membranes from peroxidative damage [10]. It relies on glutathione (GSH) as an essential co-factor. Inhibition or genetic deletion of GPX4 leads to the accumulation of L-OOH, which, in the presence of redox-active iron, propagates a destructive cascade via the Fenton reaction, culminating in ferroptosis [11]. Conversely, conditional knockout mouse models have demonstrated that GPX4 expression in various tissues is crucial for preventing ferroptosis-related damage and subsequent inflammation. The system Xc^−^ is a cystine/glutamate antiporter composed of subunits SLC3A2 and SLC7A11 [12]. It is essential for the cellular uptake of cystine, which is subsequently reduced to cysteine for the synthesis of GSH. The well-characterized ferroptosis inducer erastin inhibits system Xc^−^, depleting intracellular GSH and sensitizing cells to ferroptosis [8]. Another inducer, RSL3, directly targets and inhibits GPX4 activity [13]. The action of both inducers underscores the critical, complementary roles of system Xc^−^ and GPX4 in regulating ferroptotic cell death. Furthermore, the activity of system Xc^−^ is closely linked to the expression of SLC7A11, which can be transcriptionally regulated by factors such as ATF3.

Amino acid metabolism, iron accumulation, and lipid peroxidation related to system Xc^−^–GSH are key biochemical processes underlying ferroptosis. This will be described in the next section.

## 3. Metabolism of Ferroptosis

### 3.1. Role of Amino Acid Metabolism in Ferroptosis

System Xc^−^ is an antiporter responsible for the exchange of glutamate and cystine, playing an indispensable role in amino acid metabolism. On the extracellular side, it facilitates cystine uptake, while intracellularly, it is crucial for the synthesis of cysteine and GSH, thereby contributing to the cellular antioxidant defense system. In the 1950s, Harry Eagle’s research demonstrated that amino acid deprivation, particularly cysteine deficiency, resulted in cell death [14]. Additionally, his studies revealed that the endogenous synthesis of cysteine from methionine and glucose provided a protective effect against this cell death [15]. Cell survival depends on cysteine, which is an essential cellular antioxidant. Cysteine is also a substrate of GSH [16], and cysteine can protect cells from oxidative stress damage by promoting GSH synthesis. GSH is not only the most abundant reductant in mammalian cells, but also a cofactor of many enzymes, which can reduce the accumulation of lipid peroxides through oxidation reduction, thereby inhibiting the occurrence of ferroptosis [17]. In addition to cystine, the common amino acid metabolism also includes glutamic acid. Glutamine is the most abundant amino acid in blood and cell culture media, and its dependent performance is regulated by SLC7A11 in system Xc^−^ and has an impact on cancer [18] which once again proves the importance of amino acid metabolism for ferroptosis.

### 3.2. Role of Iron Metabolism in Ferroptosis

Iron, an essential element, exists in two primary redox states, ferrous (Fe^2+^) and ferric (Fe^3+^), and its dyshomeostasis is a pivotal driver of ferroptosis [19]. Excess cellular iron, often termed labile iron, catalyzes the generation of lethal lipid peroxides through two primary mechanisms: the Fenton reaction and by serving as a co-factor for enzymes like arachidonate lipoxygenases (ALOXs). Systemic iron homeostasis is predominantly regulated by the liver, but cellular iron metabolism is governed by key proteins including transferrin (Tf), transferrin receptor 1 (TfR1), ferritin (the primary iron storage protein), ferroportin (FPN/SLC40A1, the sole known iron exporter), and iron-responsive element-binding proteins (IRPs). Dietary Fe^3+^ is bound by circulating Tf and enters cells via TfR1-mediated endocytosis. Within the endosome, Fe^3+^ is reduced to Fe^2+^ and transported into the cytosol by divalent metal transporter 1 (DMT1). Cytosolic iron can be stored in ferritin, utilized for metabolic processes, or exported via FPN. The critical role of iron export in suppressing ferroptosis is demonstrated by evidence that overexpression of FPN ameliorates ferroptosis, while its knockdown promotes it [17]. This establishes a direct link between cellular iron handling and susceptibility to ferroptotic death [9].

The focus of this review is to explain the association between ferroptosis and immune diseases.

### 3.3. Lipid Metabolism—An Important Link in Ferroptosis

Lipid metabolism is an important link in ferroptosis, and the occurrence of ferroptosis is a death process driven by iron-dependent phospholipid peroxidation (PL), which is characterized by the accumulation of iron-dependent lethal lipid peroxide (LPO) [9]. Reactive oxygen species (ROS) are by-products of aerobic metabolism that are constantly produced, converted and consumed in all living organisms. ROS can lead to DNA damage, genetic instability and cell death by enhancing cell proliferation and survival [20]. It is well known that mammalian cells contain a certain level of PUFA-PL and bioactive iron. In the presence of bioactive iron, PUFA-PL can convert ROS into phospholipid peroxides (PLOOH) in an enzymatic or non-enzymatic manner. If PLOOH is not effectively neutralized, it will destroy the integrity of the plasma membrane and cause ferroptosis in vivo. There are many pathways to prevent lipid peroxidation, and the GPX4 pathway mentioned above is the most classic inhibitory mode, which can catalyze the reduction in toxic PLOOH to non-toxic Plol (PLOH) [21]. Moreover, GPX4 is the only enzyme that directly reduces lipid hydroperoxides in biofilms, and it can convert GSH to oxidized glutathione disulfide to reduce LPO, thus maintaining cell REDOX homeostasis [22]. In addition, there are lipophilic free radical trapping antioxidants (RTA) that can terminate the propagation of PL, thereby blocking ferroptosis caused by GPX4 deficiency [8]

In conclusion, the occurrence of ferroptosis is not a single factor, but a combination of factors. Because ferroptosis has multiple triggers, the link to the immune system is not unique. Next, we will describe the signaling pathways of ferroptosis and the immune system to further elaborate the relationship between them (Figure 1).

## 4. Ferroptosis and Signaling Pathways of the Immune System

### 4.1. The JAK/STAT Signaling Pathway

Janus kinase/signal transducer and activator of transcription (JAK/STAT) signaling pathway is considered to be one of the central communication nodes in cellular functions. Studies have found that JAK/STAT signaling pathway participates in the generation of more than 50 cytokines and growth factors, and plays an important role in immune regulation, which is closely related to various immune diseases [23].

#### 4.1.1. JAK/STAT Signaling Pathway and Immune Response

STAT transcription factors are regulated by various cytokines and growth factors and are involved in the regulation of immune responses in the microenvironment [24]. Myeloid-derived suppressor cells (MDSC) are derived from hematopoietic stem cells in bone marrow and have immunosuppressive properties of adaptive and innate immunity. A report in 2016 stated that VEGF, G-CSF, GM-CSF, Flt3L and other anti-inflammatory cytokines (such as interleukin-6) can activate STAT signaling and regulate the proliferation and activation of MDSC [25]. In particular, VEGF and IL-6 can activate STAT3 on MDSC and accelerate the proliferation of MDSC [26]. IFN-y is a key cytokine in anti-tumor host immunity, and blocking IFN-γ or destroying STAT1 will affect the inhibitory effect of MO-MDSCs [27]. Natural killer (NK) cells are important immune cells in the body and participate in the occurrence of autoimmune diseases. Proliferation of NK cells depends on IL-15 induced by STAT1 in the JAK/STAT signaling pathway [28]. In addition, STAT5 and STAT5b are important transcription factors for NK cell activation and proliferation, and the number of NK cells is significantly decreased in patients with STAT5b deficiency [29]. T cells are also associated with immune responses, and the JAK/STAT pathway plays a key role in T cells, which are the cells that produce the largest number of cellular immunities among lymphocytes. T cells can differentiate into multiple effector subsets, among which Treg cells play a regulatory role and suppress potential pathological immune responses. Treg specific transcription factor is FoxP3 [30], and its promoter binds to STAT5 to promote Treg differentiation [31]. Studies have shown that STAT3 and FoxP3 can also be used as transcription factors to regulate the biological function of Treg [32].

Activation of the JAK/STAT pathway has been found to promote the progression of various diseases, including various solid tumors [33], leukemia, inflammatory diseases, and immune diseases [34].

#### 4.1.2. JAK/STAT Signaling Pathway and Ferroptosis

IFN-γ, a key cytokine in anti-tumor host immunity mentioned above, is not only associated with the immune system, but also inhibits the transcription of Xc^−^ in erastin or RSL3-induced cell death through the JAK/STAT signaling pathway, thereby increasing cell sensitivity to ferroptosis activators [35]. In addition, it has been found that IFN-γ interacts with transcription factors in the JAK/STAT signaling pathway to affect cell ferroptosis. IFN-γ can promote the binding of STAT1 to SLC7A11, a member of the Xc^−^ family of the system, thereby increasing lipid peroxidation in vivo and slowing the growth of xenograft tumors, while STAT1 deficiency can reverse this situation [36]. Wang et al. further demonstrated that IFN-γ sensitizes adrenocortical carcinoma cells to erastin-induced ferroptosis by downregulating SLC7A11 via the JAK/STAT pathway [37] Furthermore, in hepatocellular carcinoma, IFN-γ was found to activate the JAK/STAT pathway, subsequently downregulating the mRNA and protein levels of SLC3A2 and SLC7A11, key components of system Xc^−^ [38] Beyond this, STAT family members are directly involved in regulating the expression of ferroptosis-related molecules. Propofol, a widely used anesthetic, was shown in a cellular study to induce ferroptosis in gastric cancer cells by upregulating miR-125b-5p, which suppresses STAT3 expression; this effect was reversed upon STAT3 overexpression [39]. Similarly, Zhao et al. reported that propofol could induce ferroptosis in colorectal cancer cells by modulating STAT3 [40]. As mentioned, a hallmark of ferroptosis is iron accumulation, and maintaining iron homeostasis to prevent this accumulation relies on the hormone hepcidin (corrected from ‘ferriregulin’). Research indicates that the JAK/STAT signaling pathway is a key mechanism driving hepcidin expression [41]. Iron overload subsequently promotes ROS production via the Fenton reaction, leading to cellular damage and death [38]. Certain therapeutics, such as dandelion polysaccharides in hepatocellular carcinoma [42] and auranofin (corrected from ‘Jinnophin’) in rheumatoid arthritis [43], exert their effects partly by modulating hepcidin through the JAK/STAT pathway. Conversely, ferroptosis can also regulate the STAT pathway. It has been reported that ferroptosis induction is associated with increased expression of interleukin-6 (IL-6) and interleukin-8 (IL-8). This occurs because ferroptosis activates p38 MAPK, leading to the subsequent phosphorylation and activation of STAT3, which promotes angiogenesis and upregulates inflammatory cytokine expression, thereby contributing to disease pathogenesis [44]. These studies collectively demonstrate a bidirectional crosstalk between the JAK-STAT signaling pathway and ferroptosis, highlighting potential therapeutic avenues for ferroptosis-related diseases.

### 4.2. NF-κB Signaling Pathway

#### 4.2.1. NF-κB Signaling Pathway and Immune Response

NF-κB is a classical transcription factor, and its signaling pathways are activated by different mechanisms, which can be divided into typical pathway and atypical pathway. Activation of NF-κB signaling pathway is associated with apoptosis, viral replication, tumorigenesis, inflammation and various autoimmune diseases [44]. The classical NF-κB pathway is activated by pro-inflammatory signals, Toll-like receptors (TLR) and lymphocyte receptors. TLR are pattern recognition receptors, which can recognize damage-related molecular patterns and activate the expression of related receptors and inflammatory genes, thus playing a protective role [37]. In addition, TLR is also crucial in the production of autoantibodies, and studies have found that the pathogenesis of autoimmune diseases is related to defective clearance of apoptotic cell debris [41]. Activation of the NF-κB signaling pathway is also closely related to lymphoid tissue development and function. It not only makes the development of myeloid thymic epithelial cells more clear, but also promotes the maintenance and activation of mature lymphocytes [42], which is specifically manifested as mediating T and B cell responses. The atypical pathway is mediated by NEMO and IKKβ independent IKKα dimer complexes. Recent studies have revealed that the non-classical NF-κB signaling pathway can regulate different aspects of immune function, and NIK is the core component of the non-classical NF-κB pathway [43]. The first aspect pertains to the impact on lymphocytes, which constitute a critical component of the body’s immune response. The non-canonical NF-κB signaling pathway plays an essential role in mediating the proper development of secondary lymphoid organs [45]. Research has demonstrated that the loss of NIK function in mice with lymphatic dysplasia impairs lymph node development and results in splenic structural abnormalities. Furthermore, mice deficient in NIK exhibit severe defects in the development of primary lymphoid organs, specifically the thymus. The non-canonical NF-κB pathway facilitates dendritic cells (DCs) in recognizing infections and tissue damage via pattern recognition receptors, enabling their maturation into competent antigen-presenting cells. This process is pivotal for T cell generation and activation and serves as a bridge between innate and adaptive immunity [46].

NIK serves as a central mediator in immune responses, and its abnormal activation or expression is closely associated with the onset of immune-related diseases. For instance, experiments conducted on mice and NIK knockout mice revealed B cell deficiency due to disordered lymph nodes, Peyer’s patches, and spleen structures. Additionally, NIK mutations were identified in patients with combined immunodeficiency [46]. Furthermore, individuals with NIK mutations exhibited deficiencies in follicular helper cells, memory T cell populations, and natural killer cells [46]. These findings suggest that targeting NIK could offer novel therapeutic strategies for autoimmune diseases. Highly selective and potent small-molecule inhibitors of NIK have been shown to effectively treat experimental lupus in NZB/WF1 mice [47]. Moreover, NIK plays a role in promoting inflammatory activation of human endothelial cells in the synovial fluid of rheumatoid arthritis (RA) patients, and NIK inhibitors demonstrate promising efficacy in treating RA [48]. Consistent with this, the inflammatory activation of endothelial cells induced by synovial fluid from RA patients was significantly attenuated following NIK knockdown [49]. Collectively, these results indicate that targeting NIK represents a promising approach and methodology for the treatment of immune-related disorders.

#### 4.2.2. NF-κB Signaling Pathway and Ferroptosis

NF-κB has long been recognized as a master regulator of inflammation. In a recent high-throughput mRNA sequencing experiment, we found that cystine deprivation-induced ferroptosis in smooth muscle cells was accompanied by activation of the NF-κB inflammatory signaling pathway, with sustained upregulation of p65 phosphorylation levels during ferroptosis, suggesting that ferroptosis exacerbates inflammatory responses [50] RSL3 is a well-characterized ferroptosis inducer that acts by inhibiting GPX4 activity, thereby promoting lipid peroxidation [51]. In a cellular experiment, Li et al. [52] observed that RSL3-induced ferroptosis in glioblastoma cells was characterized by elevated lipid ROS and decreased GPX4 expression. Notably, inhibition of the NF-κB pathway by BAY 11-7082 attenuated RSL3-induced ferroptosis in vitro [53]. These findings indicate a close link between NF-κB and ferroptosis. Based on this, we conducted further investigations, suggesting that modulating ferroptosis may have therapeutic potential for ameliorating certain diseases. In the classical NF-κB pathway, studies have found that miR-93-5p can promote apoptosis and ferroptosis of granulosa cells and improve polycystic ovarian syndrome (PCOS) by regulating the NF-κB signaling pathway [54]. Dimethyl fumarate (DMF) alleviates neuroinflammation and ferroptosis in chronic cerebral hypoperfusion by mediating NF-κB signaling pathway, significantly improves cognitive impairment, and partially reverses hippocampal neuronal damage and loss [55]. Heat shock protein beta-1 (HSPB1) binding to Ikβ-α and promoting its ubiquitin-mediated degradation can lead to the activation of NF-κB signal transduction, and inhibition of ferroptosis can upregulate the expression of HSPB1, thereby promoting the resistance to breast cancer treatment drugs [56]. Non-classical NF-κB activation has also been associated with ferroptosis. Specific knockout or inhibition of NIK prevented excessive lipid peroxidation in primary hepatocytes, thereby alleviating APAP-mediated hepatotoxicity in mice [57].

## 5. Ferroptosis in Immune Diseases

### 5.1. Rheumatoid Arthritis

RA is a chronic progressive inflammatory disease. Abnormal proliferation of fibroblast-like synoviocytes (FLS) drives inflammatory signals leading to the appearance of RA. RA usually affects the knee joint and elbow joint, which may lead to joint and periarticular structure damage and systemic inflammation if not treated in time [58]. Moreover, RA is characterized by the infiltration of immune cells in the joints [59], and the main clinical symptoms are joint swelling, stiffness and pain, and even bone and joint deformation and loss of function in severe cases [60]. According to the survey, the global incidence of RA is 0.5–1%, and it mostly occurs in women aged 30–50 years [61]. RA is the strongest systemic immune system disease in autoimmune diseases, and it is difficult to treat clinically and has many complications [62]. In recent years, ferroptosis has received widespread attention in the treatment of inflammatory arthritis. A bioinformatics analysis found that 34 potential ferroptosis related genes found in RA were mainly enriched in HIF-1 signaling pathway, FoxO signaling pathway, and ferroptosis pathway [63]. Most researchers hold two explanations for the role of ferroptosis in RA:

First, excessive iron accumulation can damage osteoblasts. As early as in 1996, Fritz conducted an experimental analysis of 86 synovium from patients with rheumatoid arthritis (RA) or osteoarthritis (OA) and found that iron deposits in RA synovium were significantly increased compared with OA patients [64]. Iron overload is one of the characteristics of death, and the experimental evidence that iron overload is one of the culprits of osteoporosis and bone fracture, the mice was discovered in the iron overload in mice model of [59] bone balance is broken, and in vitro experiments have confirmed that iron overload can reduce the activity of osteoblast [65]. In addition, in vitro iron was found in a mouse model of hemophilia to lead to increased expression of the p53 binding protein mdm2, which may underlie the development of hemophilia synovitis [66]. The above studies have proved that excessive iron in synovial fluid is positively correlated with the severity of RA, and iron deposition can also aggravate RA by inducing ferroptosis in macrophages. Therefore, the use of ferroptosis inhibitor (LPX-1) can alleviate the development of arthritis [67] (Table 1).

Second, lipid peroxidation can cause bone damage and aggravate immune disorders. Datta et al. measured the synovial fluid of RA patients by flow cytometry and found the existence of a large number of ROS [80]. The results of another clinical experiment showed that the levels of GSH and GPX4 in the blood of were reduced [80]. GSH can inhibit the production of ROS and the occurrence of ferroptosis, and the decreased levels indicate the increase in ROS in patients. In addition, studies have confirmed that the tumor suppressor gene p53 is expressed in RA and FSL, and p53 can inhibit systemic Xc^−^ by downregulating the expression of SLC7A11, resulting in a decrease in antioxidant capacity and ROS accumulation in the body [81]. Similarly, Mateen et al. also detected ROS production and lipid peroxidation in the blood of RA patients [82], indicating that ROS can be used as a potential marker of RA disease progression. In addition, ROS is an important element in the ROS/TNF-α feedback pathway, and the production of TNF-α depends on the activation of NF-κB signaling pathway stimulated by ROS. Studies have shown that NF-κB signaling pathway can activate p38/JNK signaling pathway to accelerate the progression of RA [83]. Other studies have shown that wasp venom can not only reduce the level of TNF-α by inactivating JAK/STAT signaling pathway, but also accumulate ROS to induce GPX4-mediated ferroptosis to treat RA [84]. The treatment of RA by inducing ferroptosis is not unique. FLS was significantly increased in a mouse model of collagen-induced arthritis, and the use of ferroptosis inducer IKE could reduce the number of fibroblasts in the synovium of mice, thereby reducing inflammation and tissue damage [68]. Glycine can reduce the expression of GPX4 and FTH1 by increasing the methylation of GPX4 promoter mediated by the concentration of S-adenosine methionine (SAM) and reducing the expression of FTH1 in RA and FLS, thereby enhancing ferroptosis and achieving the effect of RA treatment [85]. Studies have found that targeted activation of Nrf2 can not only reduce ROS, but also inhibit the proliferation and migration of FSL, indicating that inhibition of ferroptosis can also improve RA. Similarly, inhibition of NCOA4-mediated iron phagocytosis can protect RA and FLS from ferroptosis in LPS-induced inflammation under hypoxia [86] (Table 1).

### 5.2. Systemic Lupus Erythematosus

Systemic lupus erythematosus (SLE) is a typical autoimmune disease characterized by excessive activation of the immune system, resulting in lesions of autoantibodies and immune complexes, which may involve systemic tissues and organs [69]. The skin and mucosa are mainly involved, and butterfly erythema occurs, accompanied by nervous system involvement. Investigation shows that SLE is the most sex-different disease among autoimmune diseases, with a male to female incidence rate of 1:9, which is more common in young women [87]. Currently, the management of SLE remains imperfect, characterized by high treatment costs and numerous sequelae, which impose significant psychological stress and economic burdens on patients.

The accumulation of iron has been proven to be closely related to the onset of SLE [88]. Lupus nephritis (LN) is one of the most severe manifestations of lupus, and most patients will develop this condition within 5 years after the diagnosis of SLE [70]. In the kidneys of LN mouse models and in human autoimmune inflammatory diseases, the accumulation of iron increases [71,72]. Ferritin is a major iron regulator and an endogenous protective molecule against ferroptosis [88]. Yogesh et al. discovered that after establishing a lupus nephritis mouse model and treating it with hepatic ferritin, the ferritin levels in the kidneys of lupus nephritis mice significantly increased. This may help reduce iron deposition in the kidneys and the infiltration of immune cells in the kidneys, thereby further improving the condition of kidney inflammation [89]. In addition, the accumulation of ROS and lipid peroxidation play a crucial role in SLE. In the experiments of LN mice, in addition to the accumulation of iron, it was also observed that excessive ROS accumulation in renal tubular epithelial cells (RTEC) would exacerbate inflammation and fibrosis, leading to kidney damage and the progression of chronic kidney disease [90]. And using the lipid ROS inhibitor lipistat-1 can effectively inhibit the lipid ROS level in neutrophils and significantly improve the lupus model in mice [91]. Moreover, in patients with SLE, we found that lipid oxidative stress biomarkers, such as malondialdehyde (MDA), 4-hydroxy-nonenal (HNE), conjugated diene (CD), and isoprostanes, were significantly elevated and positively correlated with the disease activity of SLE [92]. These findings all confirm that iron deposition and lipid peroxidation play important roles in the development of SLE.

Neutrophils are the main immune cells in the circulation and are important in both innate and adaptive immunity [93]. Neutrophil death in SLE may serve as an autoantigen to induce interferon (IFN) production, thereby contributing to the pathogenesis of SLE [94]. Li et al. demonstrated that GPX4-induced ferroptosis of neutrophils causes the emergence of autoimmune diseases [95]. In addition, B cells also play an important role in maintaining immune homeostasis, and higher ROS levels affect the activation and differentiation process of B cells [91], while GPX4 is also essential in preventing ferroptosis of B cells [73]. In addition, studies have found that iron in T cells of SLE patients is increased compared with normal people [96], and GSH level in T cells of SLE patients is lower, and the degree of GSH reduction is related to mitochondrial hyperpolarization and increased ROS [97]. Therefore, SLE can be improved by mediating T cells, such as erucic acid inhibiting the effector function of T cells and improving the pregnancy response of SLE [74].

Kidney is one of the most severely damaged organs in SLE. Iron deposition and severe lipid peroxidation in the kidney have been observed in lupus-susceptible mouse models [98]. Iron accumulation in the kidneys of LN mice leads to albuminuria and transferrin uria [99], which can be prevented and alleviated by the ferroptosis inhibitor liproxstatin-2 [71]. In addition, the typical manifestation of SLE is skin lesions. Studies have found that the increase in skin iron content after ultraviolet B radiation exposure leads to excessive accumulation of ROS and GSH depletion, leading to the death of immunogenic keratinocytes, thereby causing skin inflammation [100] (Table 1).

### 5.3. Psoriasis

Psoriasis (PsO) is a common chronic autoimmune skin disease. The occurrence of psoriasis is related to the activation of abnormal infiltrating immune cells, excessive proliferation of keratinocytes and accumulation of inflammatory cytokines [101]. PsO is the autoimmune disease with the highest incidence at present. According to statistics, about 125 million people worldwide suffer from psoriasis, and the incidence rate is as high as 2–4% [102]. PsO clinically presents with localized or extensive erythema, papules, and desquamation, and even pruritus. According to different clinical manifestations, PsO is divided into four types, including plaque psoriasis, spotting psoriasis, erythrodermic psoriasis and pustular psoriasis, among which plaque psoriasis is the most common, accounting for 80–90% of the total incidence [103]. Unlike other autoimmune diseases, while persistent inflammation in PsO is the primary cause, genetic and environmental factors also play important roles [102]. PsO has a strong genetic susceptibility, with a prevalence of up to 17.7% in first-degree relatives of PsO, which is mainly related to alleles in the major histocompatibility complex genetic region in the short arm of chromosome 6 [104]. In addition, obesity [105], smoking [106], and bacterial infection [107] were all positively associated with the development of PsO. These external causes can trigger immune inflammatory responses in genetically predisposed patients under certain conditions. The relationship between ferroptosis and inflammatory diseases has been mentioned many times before, and PsO is no exception.

Studies have shown that the occurrence of skin diseases is mainly related to oxidative stress in the skin microenvironment, and the anti-oxidative imbalance in the external environment is the key to the pathogenesis of skin diseases [75]. The research conducted by Selvin et al. [108] indicates that for patients with long-term psoriasis, the concentrations of blood selenium in their plasma and red blood cells are significantly reduced. Selenium is a necessary trace element, and its main function is to protect the skin from harmful environmental factors. The well-known GPX4 protein is also a type of selenium protein. It is one of the main antioxidant reductases that eliminate lipid peroxidation products. Compared to healthy skin, the iron content in the skin cells of psoriasis patients increases, while the expression of GPX4 is downregulated [106]. This may be due to the fact that the low selenium content affects the synthesis of GPX4, thereby leading to the occurrence of lipid peroxidation and ferroptosis. In another study, it was also found that the GPX4 in keratinocytes of psoriasis patients was significantly reduced, and lipid peroxidation was enhanced [107]. Keratinocytes play a crucial role in psoriasis, and their death can exacerbate the inflammatory effect [76]. Moreover, gene database analysis shows that genes related to ferroptosis in psoriasis are involved in the regulation of immune microenvironment [109]. For example, acyl-coa synthetase long-chain family member 4 (ACSL4) can enhance inflammatory response by promoting lipid peroxidation and activating ferroptosis [110]. Therefore, we can infer that modulating ferroptosis is a new approach for treating psoriasis. Ferroptostain-1 (Fer-1) is an aromatic amine antioxidant and also a ferroptosis inhibitor. Studies have shown that Fer-1 can inhibit ferroptosis by blocking the production of lipid reactive oxygen species induced by Erastin, effectively treating psoriasis [76]. This might be because Fer-1 blocks the inflammatory response and alleviates skin damage by inhibiting lipid peroxidation [111]. In a mouse model of psoriasis-like skin disease induced by imiquimod, the application of Fer-1 significantly improved skin thickness and keratinization disorders [75]. These findings also provide new ideas for preventing and treating psoriasis by inhibiting ferroptosis (Table 1).

### 5.4. Inflammatory Bowel Disease

Inflammatory bowel disease (IBD) is a chronic inflammatory disorder of the gastrointestinal tract, clinically categorized into CD and ulcerative colitis (UC). CD typically manifests as widespread transmural inflammation of the gastrointestinal tract, whereas UC primarily affects the left side of the colon [112]. The most common early symptom of IBD is bloody diarrhea, with over 90% of UC patients reporting rectal bleeding [113]. Other frequent intestinal symptoms include abdominal pain, tenesmus, fecal incontinence, and vomiting [114], and in severe cases, it can lead to arthritis, liver dysfunction, and skin lesions [112]. Research has found that the pathogenesis of IBD is driven by the interaction of genetic and environmental factors. Environmental exposures such as diet, smoking, medication, and family history can disrupt the immune system [115], leading to abnormal responses of the gut microbiota [116]. Numerous studies have demonstrated the involvement of various pro-inflammatory cytokines, such as Th17, IL-1β, and IFN-γ. In UC, IL-1β has been shown to promote the development of intestinal inflammation [117]. In CD, IL-17 produced by Th17 is considered a crucial inflammatory factor in its pathogenesis, capable of activating STAT3 to induce a strong inflammatory response [118], and the inhibition of IL-17A can reduce the occurrence of inflammation [119].

Ferroptosis is a form of cell death, which plays a pivotal role in epithelial renewal, tissue homeostasis and chronic inflammation of intestinal epithelial cells [120]. Intestinal epithelial cells are highly selective barriers between the intestinal lumen and cells in the lower layer of the immune system, and stress in the endoplasmic reticulum can promote the development of chronic intestinal inflammation by downregulating tissue homeostasis [121]. Iron exposure to intestinal epithelial cells can easily trigger endoplasmic reticulum stress [122]. The involvement of ferroptosis has been widely confirmed in numerous studies. Some genes related to ferroptosis are involved in the regulation of lipid or iron metabolism, such as ACSL4, acyl-CoA synthase family member 2, GPX4 carrier SLC38A1, and glucose-6-phosphate dehydrogenase. These genes have been proven to show significant changes in ulcerative colitis specimens [77,78]. These changes led to increased iron deposition, decreased GSH levels, inactivation of GPX4, and lipid peroxidation in intestinal cells. Overall, they caused intestinal cell death and persistent inflammation, which further damaged the intestinal tissue and barrier [79,123,124].

In fact, reports of chronic inflammation of the gastrointestinal tract caused by iron exposure are not single. A Japanese study showed that either excessive dietary iron intake or oral iron treatment aggravated UC [125], and Carrier’s study also demonstrated that excessive iron aggravated intestinal inflammation [126]. Iron overload will increase the disease activity of IBD, which is due to the damage of iron overload to the intestinal antioxidant defense system, resulting in increased ROS and oxidative stress response. In addition, the ROS production detected in the colonic mucosa of IBD patients is also a strong demonstration [127]. Therefore, we reasoned that inhibition of the onset of ferroptosis could improve IBD. Curculigoside (CUR) is a natural ingredient with antioxidant and anti-inflammatory effects. The use of CUR in a mouse model of colitis can significantly upregulate the active expression of GPX4 and reduce the occurrence of ferroptosis in UC [128]. We know that GPX4 is an important antioxidant enzyme, and Dong found that by upregulating the expression of GPX4 in UC, it can significantly inhibit ferroptosis, thereby improving UC symptoms [129]. In the male Wistar rat model, the use of iron chelating agents (including maltol and kojic acid) can effectively reduce the inflammatory index [130]. These data indicate that ferroptosis inhibitors have a significant effect on IBD (Table 1).

### 5.5. Multiple Sclerosis

Multiple sclerosis (MS) is an autoimmune disease mediated by T cells, with common clinical manifestations of movement disorders, focal demyelination in the brain stem and spinal cord, which is the main cause of disability in young people [131]. In recent years, the prevalence of MS has shown an increasing trend [132], with its incidence rising as latitude increases [133]. Consequently, the etiology of MS is believed to be associated with environmental factors and geographical location [134]. Furthermore, exposure to ultraviolet B (UVB) radiation, Epstein–Barr (EB) virus infection, obesity, and smoking have been identified as potential contributors that may exacerbate the condition of MS [135]. GPX4 is broadly expressed in neurons and glial cells, where it plays a critical role in protecting these cells from oxidative stress [136]. Both GPX4 mRNA and protein levels are reduced, while lipid peroxidation is elevated in the brains of MS patients [135]. Additionally, MRI imaging has revealed an increased iron concentration in gray matter structures, providing evidence that ferroptosis is implicated in the pathogenesis of MS [137]. At present, the pathogenesis of ferroptosis in MS is not clear, and some researchers believe that oxidative stress caused by iron accumulation is one of the causes. Iron accumulation promotes neurodegeneration through pro-inflammatory mechanisms and mitochondrial dysfunction [138], and can also lead to impaired cerebral venous drainage [139].

The aforementioned studies indicate that ferroptosis inhibitors hold potential as therapeutic agents for MS. Fer-1, a commonly utilized ferroptosis inhibitor, has been shown to effectively prevent Cuprizone-induced loss of oligodendrocytes and myelin in demyelinated mice, thereby alleviating symptoms associated with MS [140]. DMF, a drug approved by the FDA for the treatment of MS, exerts its therapeutic effects by mitigating oxidative stress damage through the NRF2/NF-κB signaling pathway, thereby demonstrating significant anti-inflammatory and antioxidant properties [55,141]. Additionally, DMF has been shown to ameliorate chronic cerebral insufficiency, markedly elevate GSH levels, reduce iron expression, and alleviate hippocampal neuronal damage in rat models. Another FDA-approved agent, Desferrione (DFP), serves as a potent inhibitor of ferroptosis and is utilized in the treatment of iron overload disorders [142]. In a lysophospholipid-induced mouse model of focal demyelination in the optic nerve, DFP effectively attenuates the proliferation of microglia and astrocytes, as well as the associated myelin loss [143] (Table 1).

### 5.6. Type I Diabetes

Type 1 Diabetes Mellitus (T1DM) is also a chronic autoimmune disease characterized by pancreatic β-cell damage. According to statistical analysis of epidemiological data, T1DM is more common in adults [144], and its incidence is related to diet and living habits [145]. Research has demonstrated that diets rich in meat and protein are associated with an increased risk of developing T1DM [134,146]. This correlation is attributed to the propensity of such diets to induce hypercholesterolemia and obesity. Hypercholesterolemia, in turn, exacerbates oxidative stress, leading to the apoptosis of pancreatic β-cells [147]. Furthermore, obesity triggers a state of low-grade inflammation, wherein infiltrating macrophages release pro-inflammatory cytokines, thereby intensifying β-cell autoimmunity [148]. The pro-inflammatory cytokine IFN-γ has been shown to directly impair the function and viability of β-cells in cyclophosphamide-induced autoimmune diabetic mice [149]. The secretion of pro-inflammatory cytokines is closely associated with the activation of autoreactive T cells and the generation of ROS [150]. Moreover, lipid metabolism plays a significant role in the pathogenesis of T1DM. Free fatty acids (FFAs), which are crucial components of lipids, have been implicated in this process. Lipidomic analysis of serum FFAs in infants and young children revealed that T1DM significantly impacts the activity of lipid elongases [151]. Research has demonstrated that elevated levels of FFAs not only diminish peripheral insulin sensitivity but also contribute to β-cell dysfunction and apoptosis [152]. Long-term supplementation with ω-3 polyunsaturated fatty acids in children who have a genetic predisposition to T1DM during early stages of life can significantly reduce the likelihood of developing islet autoimmune diseases [153]. ω-3 polyunsaturated fatty acids and their bioactive derivatives have been shown to effectively suppress the inflammatory and immune responses associated with T1DM by modulating the NF-κB signaling pathway [154]. Studies have demonstrated that NaHS can effectively inhibit the release of pro-inflammatory cytokines and alleviate depression-like and anxiety-like behaviors induced by T1DM. The underlying mechanism is associated with reducing iron accumulation and oxidative stress, while increasing the expression of GPX4 and SLC7A11, thereby significantly mitigating ferroptosis in mouse models [155]. Furthermore, berberine (BBR) has been identified as a GPX4-targeting agent that effectively inhibits ferroptosis in pancreatic beta cells [156]. In addition, human umbilical cord mesenchymal stem cells (HUCMSCs) have been shown to significantly increase iron content and ROS levels in the penile tissue, leading to a notable improvement in erectile dysfunction in diabetic rats. Concurrently, HUCMSCs were found to markedly downregulate the expression of key lipid metabolism genes [157]. These findings collectively suggest a strong association between T1DM and ferroptosis, and further indicate that the inhibition of ferroptosis can ameliorate T1DM-related complications (Table 1).

## 6. Conclusions

Ferroptosis, an iron-dependent form of programmed cell death driven by lipid peroxidation, plays a dual role in autoimmune diseases: it can suppress inflammation by eliminating hyperactivated immune cells (e.g., fibroblast-like synoviocytes in RA) while exacerbating tissue damage through iron overload and lipid peroxidation (e.g., neutrophil ferroptosis in SLE). Its regulatory network involves crosstalk between signaling pathways (e.g., JAK/STAT, NF-κB) and metabolism of amino acids, iron, and lipids. For instance, IFN-γ downregulates system Xc^−^ via JAK/STAT to enhance ferroptosis sensitivity in cancer cells, whereas the NF-κB pathway influences macrophage iron metabolism through ferritinophagy. This tissue-specific dynamic regulation provides novel therapeutic targets but also poses challenges, including disease-specific mechanistic heterogeneity (e.g., protective ferroptosis in RA vs. pathogenic ferroptosis in SLE, insufficient drug selectivity, and complex metabolic interactions.

Future research must integrate interdisciplinary approaches, such as spatial transcriptomics to map ferroptosis in situ, AI-driven drug design, and humanized organoid models for translational validation. Key directions include developing tissue-targeted ferroptosis modulators (e.g., liposome-encapsulated GPX4 agonists), exploring combination therapies (e.g., ferroptosis inhibitors with anti-TNF-α monoclonal antibodies), deciphering crosstalk between ferroptosis and other cell death modalities (e.g., pyroptosis), and investigating metabolic reprogramming effects on ferroptosis susceptibility. These advancements will shift treatment strategies from “one-size-fits-all” approaches to precision regulation of the immune–metabolic–ferroptosis axis, opening new frontiers for autoimmune disease therapy.

## Figures and Tables

**Figure 1 ijms-26-10449-f001:**
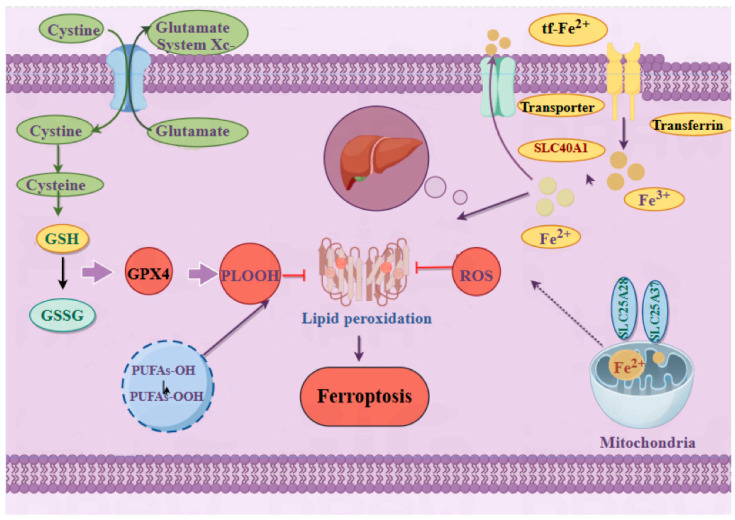
Mechanisms and signaling pathways of ferroptosis. The occurrence of ferroptosis is predominantly associated with system Xc^−^, lipid peroxidation, and iron metabolism. System Xc^−^, functioning as a cystine/glutamate antiporter, mediates the import of cystine into cells and the export of glutamate. Intracellular cystine is reduced to cysteine, which participates in the biosynthesis of GSH. GSH, under the catalytic action of GPX4, reduces lipid hydroperoxides (PLOOH)—which are formed from PUFAs—to corresponding alcohols (such as PUFAs-OH), while GSH itself is oxidized to oxidized glutathione (GSSG). In terms of iron metabolism, Tf loaded with two iron molecules interacts with the cell membrane, releasing Fe^3+^, which is subsequently reduced to Fe^2+^ via SLC40A1. Mitochondria, through transporters like SLC25A28 and SLC25A37, participate in Fe^2+^ metabolism, thereby promoting the generation of ROS. When the activity of GPX4 is inhibited or the level of GSH is insufficient, PLOOH accumulates, triggering lipid peroxidation. Meanwhile, ROS, in conjunction with Fe^2+^—related processes, exacerbates lipid peroxidation. Ultimately, the accumulation of lipid peroxidation leads to ferroptosis, and the liver may be implicated in this process. Color Key: Green: Cystine/Glutamate metabolism and Glutathione synthesis pathway. Yellow: Transferrin/iron metabolism pathway. Blue: Mitochondrial ROS production. Red: Core process of lipid peroxidation and ferroptosis.

**Table 1 ijms-26-10449-t001:** Effect of different ferroptosis treatment routes on the disease.

Disease	Approach	Mechanism	Effect	Reference
RA	By inhibiting the occurrence of ferroptosis	Mice with induced arthritis were treated with ferroptosis inhibitor (LPX-1)	Effectively relieve joint swelling and synovial hyperplasia in mice, inhibit inflammation	[68]
RA	By inhibiting the occurrence of ferroptosis	Wasp venom (WV) accumulates ROS to induce GPX4-mediated ferroptosis	Ferroptosis inducers are effective in RA treatment	[69]
RA	By inducing the occurrence of ferroptosis	For collagen-induced arthritis mice exhibiting a significant increase in fibroblast-like synoviocytes (FLS), the ferroptosis inducer IKE was administered.	Ferroptosis inducer IKE can reduce inflammation and tissue damage by reducing the number of fibroblasts in mouse synovium	[69]
RA	By inducing the occurrence of ferroptosis	Glycine was used in the CIA mouse model and the effect was evaluated	Glycine promotes ferroptosis by increasing the concentration of S-adenosylmethionine (SAM) to treat RA	[70]
RA	By inhibiting the occurrence of ferroptosis	Targeted activation of Nrf2 reduces ROS	Effectively inhibit the proliferation and migration of FSL	[71]
RA	By inhibiting the occurrence of ferroptosis	FLS isolated from RA patients were treated with LPS and ferroptosis inducers and ferroptosis inhibitors, respectively	Ferroptosis inhibitors can inhibit NCOA4-mediated iron phagocytosis to protect FLS	[72]
SLE	By inhibiting the occurrence of ferroptosis	Erucic acid was used to suppress T cells in patients with SLE	Erucic acid regulates the immune response of pathogenic T cells and improves pregnancy response in SLE	[73]
SLE	By inhibiting the occurrence of ferroptosis	Erucic acid was used to suppress T cells in patients with SLE	The ferroptosis inhibitor Liproxstatin-2 can reverse the serum induced ferroptosis in proximal renal tubular epithelial cells of LN patients and improve LN symptoms	[72,74]
PsO	By inhibiting the occurrence of ferroptosis	Fer-1 was applied to mice with IMQ-induced psoriasis-like dermatitis	Fer-1 improved the increase in skin thickness and dyskeratosis in mice	[75]
PsO	By inhibiting the occurrence of ferroptosis	The ferroptosis inhibitor Fer-1 was administered	Fer-1 inhibits lipid peroxidation to block the inflammatory response	[76]
IBD	By inhibiting the occurrence of ferroptosis	Use of CUR in a mouse model of colitis	Significantly upregulated GPX4 expression and decreased UC ferroptosis	[77]
IBD	By inhibiting the occurrence of ferroptosis	The expression of Furin protease was measured in UC	Significantly upregulated GPX4 expression and decreased UC ferroptosis	[78]
IBD	By inhibiting the occurrence of ferroptosis	Use of iron chelators (including maltol and kojic acid) in a male	Effectively reduce inflammation index	[79]

## Data Availability

The original contributions presented in the study are included in the article, further inquiries can be directed to the corresponding authors.

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
