# Peer review of "Ferroptosis in Autoimmune Diseases: Research Advances and Therapeutic Strategies"

_ijms, 2025, doi:10.3390/ijms262110449_

Round 1

Reviewer 1 Report

Comments and Suggestions for Authors

The review by He and co-workers summarizes the current state of research focusing on ferroptosis in several autoimmune diseases. While the material is comprehensive, there are several issues regarding grammar as well as indulgent use of jargon that makes the review difficult to follow. Below are some suggestions that would greatly improve the work-

Major:

  1. There is a detailed description of JAK STAT and NFkB signaling that is already know. These could be shortened and the subsequent sections for each of these (JAK-STAT and ferroptosis) and NFkB and ferroptosis need to be expanded and the exact cross-talk needs to be clarified.
  2. While autoimmune diseases is a complex problem, the exact contribution of ferroptosis and whether it is a cause or consequence of autoimmune disease needs to be explained.
  3. “Approximately 20% of the global population is impacted by over 100 distinct types of autoimmune diseases. Among them, psoriasis (2%-4%), rheumatoid arthritis (0.5%-1%), Graves' disease (0.5%), Crohn's disease (0.2%-0.3%) and systemic involvement of systemic lupus erythematosus were the most common”- needs a reference
  4. “GSH is not only the most abundant reductant in mammalian cells” – needs a reference
  5. “This was further cor- roborated by YU, who demonstrated that the down-regulation of SLC7A11 by IFN-γ through the JAK/STAT signaling pathway enhances the sensitivity of adrenocortical cells to erastin-induced ferroptosis [41].” Not clear who YU is.
  6. “For instance, experiments conducted on aly mutant mice and NIK knockout mice revealed B cell deficiency due to disordered lymph nodes, Peyer's patches, and spleen structures”. What are aly mutant mice? This is one example where a model is mentioned without any description. There are several such instances that need to be clarified and explained to a novice reader.

Minor:

  1. PUFA, NIK, APC, MHC need to be defined.
  2. The entire manuscript needs to be corrected for spacing issues, in particular between a full stop and the beginning of the next word and after or prior to references.
  3. The word ferroptosis is misspelled on page 2,4, 9 and 10.

Author Response

Thank you all for your valuable feedback! Due to the character limit of the reply box, I have compiled all the responses into a PDF file and sent it to the reviewers for their reading

Reviewer 2 Report

Comments and Suggestions for Authors

The review aims to explore the relationships between ferroptosis and the immune system in humans. It includes a description of ferroptosis and the signalling pathways of the immune system, as well as descriptions of six autoimmune diseases that report the presence of iron or ROS excess, suggesting ferroptosis. The work is unfocused, with major flaws in iron metabolism.

  • The importance of ferroptosis in the various autoimmune pathological conditions explored is not convincing, and it appears that the authors lack sufficient familiarity with iron metabolism and ferroptosis.
  • The paragraphs on ferroptosis are unclear: the reactions catalysed by GPX4 and System Xc- are not defined. The major inhibitors of ferroptosis, Erastin and RSL3, are not even mentioned. Moreover, it is not defined which are the important markers to recognise ferroptosis.
  • Similarly, for the JAK/STAT and NF-Kb pathways, the mechanism of action is not described. Although they are both involved in inflammation and ferroptosis, the relationship between the two remains undefined.
  • The role of ferroptosis in the six autoimmune disorders examined is not convincing, as in many instances, the presence of iron overload or of ROS abundance is considered an index of ferroptosis, which is not the case. Also, a reduced level of GPX4 is not a sufficient signal of ferroptosis. Some well-defined indices are necessary to recognise ferroptosis in tissues, which may include the use of specific inhibitors 
  • Ferroptosis is often misspelt as “ferroptoisi”. Moreover, the work mentions proteins such as “ferrimodulin”, “ferritransporter”, “ferriregulin, that I have never heard of before. Also, “Ferrodeath” is a new world.
  • In fig 1, top right: “transferrin with 2 iron molecules” correct: “transferrin with 2 iron atoms”. In addition, lipid peroxidation is known to occur in membranes, not in the cytosol.
  • A detailed description of the use of tofatinib and of the clinical manifestation of the autoimmune disorders seems outside the scope of the manuscript.
  • There is an excessive use of acronyms often without the corresponding open form, and this does not facilitate reading. A few sentences are ambiguous and difficult to understand

Author Response

Thank you all for your valuable feedback! Due to the character limit of the reply box, I have compiled all the responses into a PDF file and sent it to the reviewers for their reading.

Round 2

Reviewer 1 Report

Comments and Suggestions for Authors

The only issue I note is that in the revised version of the figures, the label for the transferrin receptor is hidden behind the cartoon. In needs to be moved in the front.

The authors have adequately addressed my remaining concerns. Congratulations on the publication!

Author Response

Comment:The only issue I note is that in the revised version of the figures, the label for the transferrin receptor is hidden behind the cartoon. In needs to be moved in the front.

Response:Thanks to the reviewer's suggestions, I have made the necessary changes to the pictures as requested.

Reviewer 2 Report

Comments and Suggestions for Authors

The responses to the points I raised are sufficient, but some minor problems remain.

- At the end of section 3: “The amino acid metabolism, iron accumulation, and lipid peroxidation related to system XC-GSH are the substrates for ferroptosis.”  The word substrates should be changed. Rephrase the sentence

- Section 4.2.2: “RSL3 is a well-characterized ferroptosis inducer that acts by inhibiting GPX4 activity, thereby promoting lipid peroxidation [Ref].”  add the missing reference. 

- Section 5.2 “Yogesh et al. found that after constructing ferritin in the LN mouse model,” Constructing ferritin is unclear. A transgenic mouse with a tissue specific promoter?  

- Section 5.3. “Therefore, we can infer that modulating ferroptosis is a new approach for treating psoriasis. Iron-1 is an aromatic amine antioxidant and also a ferroptosis inhibitor.” The inhibitor Iron-1 is unknown to me and not present in refs 110 and 111.  

Author Response

Comments1:At the end of section 3: “The amino acid metabolism, iron accumulation, and lipid peroxidation related to system XC-GSH are the substrates for ferroptosis.”  The word substrates should be changed. Rephrase the sentence

Response1: We sincerely thank the reviewer for this insightful suggestion.We have revised this according to your suggestions, and the revised content is as followsAmino acid metabolism, iron accumulation, and lipid peroxidation related to system Xc⁻GSH are key biochemical processes underlying ferroptosis(Page3)

Comments2:- Section 4.2.2: “RSL3 is a well-characterized ferroptosis inducer that acts by inhibiting GPX4 activity, thereby promoting lipid peroxidation [Ref].”  add the missing reference. 

Response:We thank the reviewer for raising this critical point. We have supplemented the relevant literature as followsJ. Wang et al., "Structural optimization and biological evaluation of ferrocene-appended RSL3 derivatives as potent ferroptosis inducers," (in eng), Eur J Med Chem, vol. 301, p. 118162, Sep 16 2025, doi: 10.1016/j.ejmech.2025.118162.(Page8)

Comments3:Section 5.2 “Yogesh et al. found that after constructing ferritin in the LN mouse model,” Constructing ferritin is unclear. A transgenic mouse with a tissue specific promoter?  

Respond:We appreciate the reviewer's valuable comments. In light of these comments, we have revisited the literature and revised this sentence. The revised content is as follows:Yogesh et al. discovered that after establishing a lupus nephritis mouse model and treating it with hepatic ferritin, the ferritin levels in the kidneys of lupus nephritis mice significantly increased. This may help reduce iron deposition in the kidneys and the infiltration of immune cells in the kidneys, thereby further improving the condition of kidney inflammation(Page11)

Comments4:“Therefore, we can infer that modulating ferroptosis is a new approach for treating psoriasis. Iron-1 is an aromatic amine antioxidant and also a ferroptosis inhibitor.” The inhibitor Iron-1 is unknown to me and not present in refs 110 and 111.

Respond:We appreciate the reviewer's suggestions. After rechecking the relevant literature, we have deleted the erroneous parts of this content and revised all instances of "Iron-1" to "Fer-1" throughout the manuscript.(Page12)
